# Understanding Vocational Students’ Motivation for Dietary and Physical Activity Behaviors

**DOI:** 10.3390/ijerph18041381

**Published:** 2021-02-03

**Authors:** Annabelle Kuipers, Gitte C. Kloek, Sanne I. de Vries

**Affiliations:** 1Research Group Healthy Lifestyle in a Supporting Environment, The Hague University of Applied Sciences, Johanna Westerdijkplein 75, 2521 EN The Hague, The Netherlands; annabellekuipers@hotmail.com (A.K.); s.i.devries@hhs.nl (S.I.d.V.); 2Faculty of Earth and Life Sciences, VU University Amsterdam, De Boelelaan 1105, 1081 HV Amsterdam, The Netherlands

**Keywords:** motivation, self-determination theory, diet, physical activity, vocational students

## Abstract

Unhealthy eating behaviors and low levels of physical activity are major problems in adolescents and young adults in vocational education. To develop effective intervention programs, more research is needed to understand how different types of motivation contribute to health behaviors. In the present study, Self-Determination Theory is used to examine how motivation contributes to dietary and physical activity behaviors in vocational students. This cross-sectional study included 809 students (mean age 17.8 ± 1.9 years) attending vocational education in the Netherlands. Linear multilevel regression analyses were used to investigate the association between types of motivation and dietary and physical activity behaviors. Amotivation was negatively associated with breakfast frequency and positively associated with diet soda consumption and high-calorie between-meal snacks. A positive association was found between autonomous motivation and water intake, breakfast frequency, fruit intake, and moderate-to-vigorous physical activity. Autonomous motivation was negatively associated with the consumption of unhealthy products. Controlled motivation was not associated with physical activity or dietary behaviors. Different types of motivation seem to explain either healthy or unhealthy dietary behaviors in vocational students. Autonomous motivation, in particular, was shown to be associated with healthy behaviors and could therefore be a valuable intervention target.

## 1. Introduction

Unhealthy dietary behaviors and a decrease in physical activity are a major problem in adolescents and young adults [1]. These unhealthy behaviors may lead to a higher risk of non-communicable diseases later in their lives [2,3,4,5]. The prevalence of these unhealthy behaviors is not evenly distributed amongst different groups of young people [6,7]. Students attending vocational education have less favorable health practices than students attending higher secondary education [8]. In the Netherlands, many vocational students are overweight, show more sedentary behavior than recommended, do not engage in sufficient physical activity, and most do not meet the guidelines for fruit and vegetable consumption [9,10,11]. The vocational education and training (VET) sector is regarded as critical to the Dutch economy because about 40% of the working population has obtained a vocational qualification [12]. The acquisition of citizenship skills that refer to the willingness and ability to reflect on one’s lifestyle and care for one’s vitality as a citizen and employee are a key competence in the Netherlands and considered of great value in the future employability of Dutch vocational students [13].

Most vocational students are in their late adolescence, ranging in age from 16 to 24 years old. Late adolescence is characterized by many cognitive changes, including changes in motivational processes [14,15,16]. It is also a period of transition from adolescence to adulthood in which young people establish independence, adopt lasting health behavior patterns, and are at high risk of developing obesity and unhealthy dietary habits and physical activity practices [17]. Therefore, it is highly relevant to promote healthy dietary and physical activity behaviors among adolescents and young adults, especially among those belonging to a vulnerable target group.

To develop effective intervention programs for vocational students, more research is needed to identify factors that predict the engagement in health behaviors among this population. A wide range of theoretical explanations provides a basis for understanding the determinants of behavior and behavior change [18]. Several studies have reported that successful behavior change maintenance depends on motives, self-regulation, resources, habits, and environmental and social influences, and point out that motivation is a critical factor in supporting healthy dietary and physical activity behaviors [19,20,21,22]. As a general theory of motivation, the Self-Determination Theory (SDT) is being applied to study motivation in numerous health care and health promotion contexts [23,24]. SDT distinguishes different types of motivation on a continuum in terms of the degree to which the motivation is (non)self-determined: amotivation, controlled motivation, and autonomous motivation [25]. Amotivation is a type of motivation in which an individual does not have any intention to perform a certain behavior. Autonomous motivation describes a self-determined type of motivation. In this type of motivation, behavior is performed for the individual’s sake and the goal is self-satisfaction. The motivation type between amotivation and autonomous motivation is controlled motivation, where the motivation to act is driven by external factors and is less self-determined than autonomous motivation. Driving factors can include social influences such as friends and family, or teachers [25].

In early and mid-adolescence, high autonomous motivation is known to be related to increased fruit and vegetable intake [26]. Furthermore, in Finnish vocational students aged 17–19 years, a higher level of autonomous motivation was related to increased physical activity [27]. Similar results have been found in middle-school students aged 12 years and in children and adolescents aged up to 18 years old [28,29,30]. Controlled motivation, on the other hand, shows a weak negative association with physical activity [30]. Moreover, individuals who report amotivation to live healthily have poor uptake and adherence to health behaviors. They do not see any reason to change their behavior and are not likely to implement any changes [31].

The evidence above shows that types of motivation influence the engagement in health behaviors in (young) adults and adolescents. However, most research on adolescents is focused on early or mid-adolescence. There is little research on the motivation to make healthy choices of late adolescents, particularly vocational students. Therefore, the purpose of this study is to examine the association between type of motivation and dietary and physical activity behaviors among adolescents and young adults in vocational education.

## 2. Methods

### 2.1. Design

In this cross-sectional study, data was collected as part of a peer-delivered, school-based healthy lifestyle intervention aimed at improving physical activity and dietary behaviors among vocational students, between November 2017 and January 2018. The study was not within the scope of the Dutch Medical Research Involving Human Subjects Act [32] and was therefore exempt from review by the Medical Ethics Committee Southwest Holland.

### 2.2. Study Population

The study population consisted of a convenience sample of 809 vocational students studying at three VET school locations in a metropolitan area of the Netherlands. The following steps were taken to include students in the study. First, program education managers of seven different VET programs were contacted by the researchers. Based on the program schedule, the managers selected as many first- and second-year classes as possible during a school week. All students in the selected classes were eligible to participate. Next, students attending the selected classes were informed that they could voluntarily participate in the study and that recruitment would take place during class. In class, they received oral and written information from the researchers before they had to decide to participate. After they gave their consent to participate through a digital informed consent form, students completed an online survey. 

### 2.3. Measurements

#### 2.3.1. SDT Motivation Types

For both diet and physical activity motivation, the Treatment Self-Regulation Questionnaire (TSRQ) was used to determine the participants’ motivation to either eat healthily or exercise regularly. A validation study showed adequate internal consistency (α = 0.73–0.93) for all three subscales: autonomous motivation, controlled motivation, and amotivation. In total, the TSRQ consists of 15 statements: six on autonomous motivation, six on controlled motivation, and three on amotivation [33]. A translation of the TSRQ statements was available in Dutch and is widely used in various studies about the motivation for health behavior, but a Dutch version was never formally validated. In this study, the TSRQ either began with the stem, ‘The reason I would eat a healthy diet is:’ or ‘The reason I would increase my physical activity level is:’ followed by 15 statements that represent a reason for engaging in the health behavior mentioned. Participants were asked to rate how true each statement was for them using a 5-point scale. For example, one of the statements for autonomous motivation, ‘I would eat healthier because I feel responsible for my health’, could be scored from 1 (not at all true) to 5 (very true). For each subscale, the scores on the statements were summed and divided by the number of statements belonging to the scale to calculate a composite score (range 1–5) for all three subscales. A higher score on the scale reflects a stronger reason to engage in the health behavior from the perspective of autonomous or controlled motivation or a stronger reason to not engage in the health behavior in case of amotivation.

#### 2.3.2. Dietary Behavior

We chose dietary behaviors that frequently occur in the school environment. To assess fruit, breakfast, sweet and savory snacks, water, energy drink, and soda consumption, existing questions (some were validated) from the periodic youth health monitor system of the Dutch regional health authorities were used [34]. An example question is ‘On how many days a week do you consume fruit?’. Eight options were given, from never to seven days a week. After this, we asked ‘How much fruit do you generally eat on those days?’. Answer options included 0.5, 1, 1.5, 2, 2.5, and 3 or more pieces of fruit per day. To calculate the mean daily fruit consumption, the number of days was multiplied by the amount of fruit consumed and divided by seven. A similar calculation was done for the other dietary behaviors, taking into account the frequency of the behavior and the amount consumed. Furthermore, results were compared with the dietary guidelines of the Netherlands Nutrition Centre [35]. The guidelines for dietary behaviors are unsweetened fluid intake (1.5 L/day), fruit consumption (2 pieces/day), diet soda (a maximum of 4 units of 250 mL/day), breakfast consumption on all days of the week, and a maximum of 3 high-calorie snacks a week [35]. To calculate the number of high-calorie snacks, sugar-sweetened beverages and sweet and savory snacks were summed. 

#### 2.3.3. Physical Activity

Physical activity level was assessed using the validated SQUASH Short QUestionnaire to ASsess Health-enhancing physical activity (Spearman correlation coefficient for reproducibility = 0.58 and relative validity = 0.45), which measures walking and cycling to school or work, activity at work, household tasks, activity in free time, and sports [36]. Participants were asked to report the frequency (days/week) and duration (minutes/day) in which they engaged in these activities during a normal week. It was necessary to adapt the wording of some questions for the participants to ensure clarity. Physical activity during work was, for example, changed to physical activity during a side job. The results were converted to minutes per week spent in moderate- or vigorous-intensity physical activity and a sum score of moderate and vigorous physical activity (MVPA) together based on Metabolic Equivalent Tasks (METs) derived from Ainsworth’s compendium of physical activity [36,37]. Physical activity levels were compared with the physical activity guidelines of the Health Council of the Netherlands [38]. The guidelines for physical activity behaviors are 150 min MVPA/week for adults and 60 min MVPA/day for youth younger than 18 years [38].

#### 2.3.4. Covariates

Covariates in this study included gender, age, weight status, and training level. Previous research has suggested that gender, age, and weight status influence diet and physical activity behaviors [39,40,41]. Gender was determined as male or female. The participants’ birth year was used to determine their age. Their Body Mass Index (BMI) was calculated based on self-reported height and weight and the international IOTF cut-off points were used to determine their weight status [42]. Because the general education level is associated with a healthier lifestyle [43], it was decided to correct for training level as well. Vocational education in the Netherlands has four different qualification levels [12], level one being the lowest and four the highest. The levels were combined into three groups: level 1 & 2, level 3, and level 4. 

#### 2.3.5. Statistical Analyses

The analyses included descriptive statistics of the participants. Outliers and missing data were explored and, if needed, participants were removed from the analysis. 

Due to the hierarchical structure of the data, with 809 students (individual level) clustered within seven VET programs (group level), we used linear multilevel regression analyses to investigate the associations between type of motivation and physical activity or dietary behaviors. For all dependent variables, a three-step modeling strategy was employed. For the null model, the first step comprised the calculation of the Intraclass Correlation Coefficient (ICC) to identify the proportion of the variance in the dependent variable (behavior) explained by the grouping structure of the population (VET program). The second step involved the inclusion of the type of motivation variables. The covariates were added in the third step to produce the final models. To investigate the associations between type of motivation and physical activity or dietary behavior, the calculated regression coefficients and 95% confidence intervals (CI) were used. For the dietary behaviors, we only fitted models for the behaviors included in the Dutch dietary guidelines [35]. An alpha of 0.05 was used to test statistical significance. All analyses were conducted using IBM SPSS version 26.

## 3. Results

The study population characteristics can be found in Table 1. The mean age of the study population was 17.8 (±1.9) years old. The majority of the sample had a normal weight (75%) and was female (62%). The type of VET program varied, but a large part of the sample followed the Lifestyle & Sports program (31%). All training levels were represented, but the majority of participants (66%) attended the highest level, level 4.

In Table 2, descriptive statistics for dietary and physical activity behaviors can be found. The mean water consumption was 1086 mL/day, the mean soda consumption was 330 mL/day, and the mean consumption for diet soda was 133 mL/day. For fruit, the mean consumption was 0.9 pieces a day and the mean number of high-calorie snacks was 1.9 portions per day. Breakfast frequency was, on average, 4.7 days per week. Almost all participants met the guideline for diet soda, but only 12% met the guideline for fruit consumption. In addition, only a limited percentage of participants met the guideline for a maximum of three high-calorie snacks per week. 

The mean time for weekly MVPA was 935 min, 161 min for vigorous physical activity, and 774 min for moderate physical activity. Overall, 49% met the guideline for MVPA. However, stratification by age group shows that 20% of participants younger than 18 years met the recommended level of MVPA (at least 60 min MVPA per day), while 79% of participants aged 18 years and over met the adult MVPA guideline of at least 150 min MVPA per week.

The median motivation scores are highest for autonomous motivation for both diet and physical activity (Table 3), followed by controlled motivation and lastly amotivation. Controlled motivation shows higher values for physical activity than it does for diet. Apart from this, scores are similar for diet and physical activity.

Table 4 and Table 5 show the associations of autonomous motivation, controlled motivation, and amotivation with physical activity and dietary behaviors, as determined by multilevel linear regression analysis. For dietary behavior (Table 4), the final models showed an association between autonomous motivation and all dietary variables, except for diet soda. A negative association between autonomous motivation and amount of high-calorie snacks can be seen, meaning that with every increase of 1 in autonomous motivation score, 3.9 fewer high-calorie snacks are consumed. Autonomous motivation also showed a positive association with the fruit and water intake per day and the number of days in which breakfast was consumed. Controlled motivation showed no significant associations with any of the dietary variables. For amotivation, positive associations were found with the portions of high-calorie snacks consumed per week and with diet soda consumption, while a negative association was found with the number of days in which breakfast was consumed. The addition of covariates to the final models had little effect on the associations found in the unadjusted models. The exception was water intake, where the association with amotivation was no longer significant after adjustment for covariates. In the null models, the grouping structure of the population by VET program explained a small part of the variance in the dietary behaviors. The percentage of variance explained ranged from 0.1% for water intake to 8.6% for breakfast consumption.

For physical activity behavior (Table 5), autonomous motivation was positively associated with the number of minutes per week MVPA in the final model. No other significant associations with motivation were found in the unadjusted or final models. In the null models, the grouping structure of the population by VET program explained a small part of the variance in the physical activity. The percentage of variance explained ranged from 0.7% for moderate physical activity to 9.5% for vigorous physical activity.

## 4. Discussion

This study aimed to investigate the association between type of SDT motivation and diet and physical activity behaviors. A considerable number of vocational students do not comply with the dietary guidelines or physical activity recommendations. At the individual level, the type of motivation partly explained their dietary and physical activity behaviors. 

Autonomous motivation was associated with consuming fewer unhealthy products per week and consuming more water, breakfast frequency, fruit consumption, and conducting more MVPA. For moderate and vigorous physical activity separately, no motivation type seems to be of influence. It seems that the association between autonomous motivation and physical activity concerns only the sum of moderate and vigorous physical activity. Autonomous motivation thus seems to explain most of the healthy dietary behaviors and physical activity behavior of vocational students. In previous studies, autonomous motivation is described as the type of motivation that facilitates persistence and sustainability of behavior due to its high levels of autonomy, while controlled motivation does not lead to sustainable behavior [25,44,45]. This explains why autonomous motivation is important in healthy dietary behaviors and physical activity behavior in vocational students. Results of other studies seem to follow the same pattern: in early and mid-adolescents, autonomous motivation was found to be associated with increased fruit intake and increased physical activity [26,27,28,30], and in vocational students, autonomous motivation was found to be associated with MVPA [27].

Controlled motivation did not show an association with any variables: it does not appear to be enough to maintain the healthy lifestyle behaviors investigated in this study. It may not lead to sustainable behavior, because controlled motivation is characterized by lower levels of autonomy compared with autonomous motivation. This could lead to a relapse into old behavior, as the external factor that drove the controlled motivation is removed over time [46].

Amotivation was associated with consuming more unhealthy products per week and consuming breakfast less often. This type of motivation seems to be associated with unhealthy dietary behaviors. A possible explanation for this negative association is the indifferent attitude that vocational students have toward making healthy lifestyle choices. Giles and Brennan [47] found that British late adolescents (aged 18–25) are not willing to put much effort into adopting a healthy lifestyle and thus have a rather indifferent attitude to it. This seems to be the same in Dutch vocational students. If this indifferent attitude leads to amotivation, it could explain its negative effect on dietary behavior. Amotivation, however, also showed a significant positive effect on the consumption of diet soda. A possible explanation for this is the fact that such students show no awareness of calorie content in beverages. It was found that the most important factors for choosing beverages for college students (mean age 19 years) were taste and price [48]. Therefore, health might not be an important factor for vocational students when consuming diet soda.

At the group level, the type of VET program did not explain a large part of the variance in the dietary and physical activity behaviors. It was expected that program type was an important factor of clustering in the data because the social norm is very important for young adults [47]. The low ICC could be because the group-level variable used to adjust for clustering of the data might have been too heterogeneous. Social norms may have more influence in more homogeneous groups, such as class level instead of the type of VET program.

### 4.1. Limitations, Strengths, and Recommendations 

The first limitation of this study is the use of self-administered questionnaires. This may have caused recall bias. In the case of diet and physical activity questions, participants tend to be too positive about their habits [49]. In this study, recall bias could have led to an overestimation of the diet and physical activity behavior of the study population. Additionally, the SQUASH questionnaire is known to overestimate the physical activity that participants conduct, which could have caused an overestimation of the physical activity of vocational students [50]. The exact effect that these possible overestimations might have had on the found associations cannot be inferred. Secondly, the cross-sectional nature of this study is a limitation as the type of motivation and behavior were measured at the same time, so their interrelationship does not necessarily reflect a causal relationship. In absence of a time dimension, for example, it is not possible to determine whether higher scores on autonomous motivation precede healthier dietary behaviors. 

Finally, the study population consisted of vocational students from three VET school locations in the metropolitan area of the Netherlands. This makes the results not automatically applicable to VET programs in general. Female students were overrepresented in the sample, which may have caused an overestimation of the diet and physical activity behavior of the study population because being female is related to having healthier lifestyle habits [41]. Furthermore, the sample included a large number of Lifestyle & Sports students. This type of VET program attracts students who are interested in sports and lifestyle. Therefore, this could have caused an overestimation of the diet and physical activity behavior of the study population, especially in the amount of physical activity vocational students engage in. The effects that the abovementioned factors had on the associations cannot be inferred. The external validity of this study could thus be improved by obtaining a more representative sample of vocational students.

Despite the abovementioned limitations, to our knowledge, this study is one of the first that reports associations between type of SDT motivation and dietary and physical activity behavior of vocational students. Therefore, it provides new and much-needed insights into their motivation and health behavior. Moreover, the large sample size of the study increased its reliability. Furthermore, the use of multilevel analyses strengthened the study’s conclusions because variability due to clustering of the data was accounted for. 

For future research, we recommend diving deeper into the topic of SDT and self-directed health behaviors among vocational students. More insight is needed into the three basic psychological needs—autonomy, competence, and relatedness—and their relationship with autonomous motivation and amotivation, to develop health-promoting interventions for this group. In addition to diet and physical activity behavior, more variables can be investigated to get a more complete picture of the determinants of vocational students’ health behavior. For example, we chose dietary behaviors that are frequent in the school environment, but other dietary behaviors that are more common in the students’ home environment (such as vegetable consumption) are also interesting to investigate further.

### 4.2. Implications 

The results of this study show a clear association of autonomous motivation with dietary behavior and MVPA in vocational students. This implies that autonomous motivation is a reasonable target in the development of health-promoting interventions. 

A review by Ng et al. [23] showed that enhancing autonomous motivation led to beneficial health outcomes. Furthermore, satisfying basic psychological needs was found to be important. To enhance autonomous motivation, autonomy-supportive interventions must focus on four SDT components. First, they must increase the sense of competence of participants. Second, these feelings of competence must be coupled with feelings of autonomy. Third, interventions must make sure to give participants a sense of security or relatedness. Lastly, extrinsic rewards must be avoided as they stimulate controlled motivation instead of autonomous motivation [46]. One possible intervention to enhance autonomous motivation is motivational interviewing, as this is a method to adhere to behavior change with many parallels with the mentioned SDT concepts [51]. In adolescents, motivational interviewing was found to be effective in promoting several healthy behaviors [52]. In addition, peer relations have a positive effect on autonomous motivation. In adults with weight management goals, for example, it was found that autonomy support by significant others led to satisfaction of psychological needs, which is beneficial for autonomous motivation [53]. Gairns et al. [54] found that high school students showed stronger autonomous motivation for participation in a physical education class when they felt a positive relatedness with their classmates. In the context of VET qualification, Lifestyle & Sports students, for example, may learn how to motivate a target population such as their fellow students in other VET programs to improve health-related behaviors. In this way, they may act as a significant other but also improve the competences they need as future professionals.

## 5. Conclusions

In general, dietary and physical activity behaviors of vocational students are poor. On the one hand, autonomous motivation is associated with their healthy diet and physical activity behaviors. On the other hand, amotivation shows associations with unhealthy dietary behaviors of vocational students. Controlled motivation does not show any associations with their diet and physical activity behavior. Because of its positive association with a healthy diet and physical activity behavior, autonomous motivation seems to be a valuable target for new, autonomy-supportive interventions to improve the healthy lifestyle of vocational students.

## Figures and Tables

**Table 1 ijerph-18-01381-t001:** Individual characteristics as a percentage of the sample (N = 809) or means and standard deviations.

Characteristic	*n* (%)	Mean (SD)
Age (years)		17.8 (1.9)
BMI (kg/m^2^)		22.3 (3.5)
Weight status		
Underweight	56 (7)	
Normal weight	603 (75)	
Overweight	124 (15)	
Obese	26 (3)	
Gender		
Male	303 (38)	
Female	506 (62)	
Type of VET program		
Economics & Law	162 (20)	
IT	20 (3)	
Lifestyle & Sports	249 (31)	
Social Work	66 (8)	
Beautician	161 (20)	
Health Care	60 (7)	
Fashion	91 (11)	
Training level		
Level 1 & 2	89 (11)	
Level 3	188 (23)	
Level 4	532 (66)	

**Table 2 ijerph-18-01381-t002:** Summary of means and standard deviations, median, and interquartile range for dietary and physical activity behaviors and percentage of the sample (N = 809) that met the guidelines.

Behaviors	Mean (SD)	Median (IQR)	% Guideline *
Water (mL/day)	1086 (671)	1000 (536–1500)	35
Regular soda (mL/day)	330 (431)	143 (0–500)	na
Diet soda (mL/day)	133 (294)	0 (0–107)	98
Energy drink (mL/day)	65 (143)	0 (0–71)	na
Fruits (pieces/day)	0.9 (2.6)	0.7 (0.3–1.3)	12
High-calorie snacks (portions/day)	1.9 (1.8)	1.4 (0.7–2.6)	8
Breakfast (days/week)	4.7 (2.6)	6.0 (3.0–7.0)	48
MVPA (minutes/week) **	935 (708)	840 (420–1260)	49
Moderate PA (minutes/week) **	774 (646)	690 (270–1080)	na
Vigorous PA (minutes/week) **	161 (305)	0 (0–270)	na

* Percentage that meets the guidelines. ** For physical activity (PA) behaviors, 2 participants were excluded from the analysis due to unrealistically high scores per week. IQR Interquartile Range; na not applicable.

**Table 3 ijerph-18-01381-t003:** Summary of means and standard deviations, median, and interquartile range for scores on different types of motivation for diet and physical activity.

Type of Motivation	Mean (SD)	Median (IQR)
Autonomous motivation diet	3.7 (0.8)	3.7 (3.2–4.2)
Controlled motivation diet	2.4 (0.7)	2.4 (2.8–3.8)
Amotivation diet	2.2 (0.7)	2.2 (1.7–2.7)
Autonomous motivation physical activity	3.7 (0.8)	3.7 (3.2–4.2)
Controlled motivation physical activity	2.5 (0.8)	2.5 (2.0–3.0)
Amotivation physical activity	2.0 (0.7)	2.0 (1.3–2.7)

**Table 4 ijerph-18-01381-t004:** Multilevel linear regression analyses of the association between types of motivation and dietary behaviors (N = 809).

Variable	Null Model	Unadjusted Model ^a^		Final Model ^b^	
		Beta	95% CI	Beta	95% CI
High-calorie snacks (portions/week)					
Autonomous motivation score diet		−3.8 **	−6.1 to −1.5	−3.9 **	−6.1 to −1.6
Controlled motivation score diet		−1.8	−4.1 to −0.4	−1.6	−3.8–0.7
Amotivation score diet		5.7 ***	3.6–7.9	4.5 ***	2.3–6.7
ICC level 2	0.035	0.022		0.038	
Diet soda (mL/day)					
Autonomous motivation score diet		14	−19 to −48	13	−20–47
Controlled motivation score diet		15	−18–48	15	−18–48
Amotivation score diet		40 *	8.6–72	33 *	0.3–65
ICC level 2	0.029	0.026		0.022	
Fruit (pieces/day)					
Autonomous motivation score diet		0.2 ***	0.1–0.3	0.2 ***	0.1–0.3
Controlled motivation score diet		0.0	−0.1–0.1	0.0	−0.1–0.1
Amotivation score diet		0.0	−0.1–0.1	0.0	−0.1–0.1
ICC level 2	0.018	0.021		0.022	
Breakfast (times/week)					
Autonomous motivation score diet		0.5 **	0.2–0.8	0.5 **	0.2–0.8
Controlled motivation score diet		−0.1	−0.4–0.2	−0.1	−0.4–0.2
Amotivation score diet		−0.3 *	−0.6 to −0.1	−0.3 *	−0.6 to −0.1
ICC level 2	0.086	0.082		0.071	
Water intake (mL/day)					
Autonomous motivation score diet		158 ***	84–232	164 ***	90–237
Controlled motivation score diet		55	−19–129	46	−28–120
Amotivation score diet		−72 *	−142 to −1.2	−66	−138–5.9
ICC level 2	0.001	<0.001		0.001	

^a^. outcome variables were analyzed separately, motivation variables were entered simultaneously in each model. ^b^. values were adjusted for the following covariates that were entered simultaneously in each model: gender (male/female), age, weight status (underweight, normal weight, overweight, and obese), and training level (level 1 & 2, 3, and 4). * significant value α < 0.05, ** significant value α < 0.01, *** significant value α < 0.001. ICC Intraclass Correlation Coefficient.

**Table 5 ijerph-18-01381-t005:** Multilevel linear regression analysis of the associations between type of motivation and physical activity (N = 807 ^a^).

Variable	Null Model	Unadjusted Model ^b^		Final Model ^c^	
		Beta	95% CI	Beta	95% CI
Moderate PA (min/week)					
Autonomous motivation score PA		49	−17–115	56	−10–122
Controlled motivation score PA		−19	−91–53	−23	−95–49
Amotivation score PA		−25	−95–45	−29	−99–42
ICC level 2	0.007	0.007		0.007	
Vigorous PA (min/week)					
Autonomous motivation score PA		20	−8.4–48	19	−9.4–47
Controlled motivation score PA		21	−10–51	22	−8.7–52
Amotivation score PA		−12	−42–18	−19	−49–11
ICC level 2	0.095	0.135		0.095	
MVPA (min/week)					
Autonomous motivation score PA		69	−2.6–140	74 *	2.4–146
Controlled motivation score PA		8.0	−70–86	6.4	−72–84
Amotivation score PA		−40	−116–36	−51	−128–25
ICC level 2	0.014	0.027		0.014	

^a^. two participants were excluded from analysis due to unrealistically high minutes of physical activity (PA) per week. ^b^. outcome variables were analyzed separately, motivation variables were entered simultaneously in each model. ^c^. values were adjusted for the following confounders that were entered simultaneously in each model: gender (male/female), age, weight status (underweight, normal weight, overweight, and obese), and training level (level 1 & 2, 3, and 4). * significant value α < 0.05.

## Data Availability

The data presented in this study are available on request from the corresponding author. The data are not publicly available due to to the original consent not containing approval for public data sharing.

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
