# Peer review of "Understanding Vocational Students’ Motivation for Dietary and Physical Activity Behaviors"

_ijerph, 2021, doi:10.3390/ijerph18041381_

Round 1

Reviewer 1 Report

  1. This topic is interesting and provides more useful information on the dietary and physical activity behaviors.
  2. The author took the vocational students as the study sample, and focused on the young stage of the adults. I suggest the author should provide more explanations to consider them as the sample from the viewpoints of vocational education or training analysis.
  3. The author employed the reasonable and appropriate statistical method to analyze the sample data. I suggest the author could provide some information about the validity and reliability of the measurement tools.
  4. The author found the relationship among autonomous motivation, diet and physical activity behaviors. I suggest the author could focus on these results to provide some useful strategies for the vocational education or workplace trainings.

Reviewer 2 Report

Journal: Int. J. Environ. Res. Public Health

Manuscript Number: 1040634-peer-review-v1

Type of manuscript: Original Article

Title: Understanding vocational students’ motivation for dietary and physical activity behaviors

GENERAL COMMENTS

The present study assesses motivation factors interfering with diet and physical activity in a Dutch cohort of 809 students.

It appears like a novelty due to the target population, and this would make it interesting.

In my opinion, the introduction and discussion should be revised and tightened; details are missing in the methods section, and results could be more precisely explained.

Please see detailed comments.

ABSTRACT

A discussion or conclusion section in the abstract is missing.

INTRODUCTION

  1. In my opinion, the introduction could be revised, avoiding redundant information and repetition.
  2. Line 29: in my opinion, “decline” is not the correct word referring to physical activity.
  3. Line 32, “not evenly distributed across young people”: after this sentence, I expect you to speak about adults or older people. As you explain later, vocational students are still young people or young adults (age 16-24). Please, reformulate the sentence.
  4. Lines 62-74: in my opinion, this paragraph could be abbreviated, and previous studies could be explained in depth in the discussion section. In this way, the reader can contextualize your study results with prior knowledge.

MATERIALS AND METHODS

  1. Line 89: what do you mean by “convenience study”?
  2. In which city did the recruitment take place?
  3. Did you obtain any written consensus from the parents of minors students?
  4. SDT: which is the result’s range? How is the final result interpreted? Higher scores correspond to..? In the end, is it possible to obtain three different values referring to three motivation types?
  5. Are these three questionnaires Dutch validated?
  6. Lines 102-103: what does this sentence mean?
  7. Lines 130-131: is this procedure already validated or described in another study?
  8. Lines 139-140: why did you decide to adjust for training level if education level is associated with a healthier lifestyle?
  9. Statistical analysis: did you check for normality, skewness, or kurtosis? Why within seven vocational education training program (line 147)?
  10. Some statistical methods used after are missing: percentage, quartiles.

RESULTS

  1. What is VET? And VET levels?
  2. In my opinion, numbers should be reported (not only percentages).
  3. Lines 165-179: please follow the order of table 2 to discuss your results.
  4. Lines 168-169: why did you consider together these dietary products? It is not easy to understand how did you obtain this result.
  5. What is PA?
  6. Please, make it more transparent that MVPA is the sum of the other two physical activity typologies.
  7. Lines 176-179: How did you make a classification by age? Which is the cut-off value, and how did you obtain it? You did not report this analysis in the statistical section.
  8. In my opinion, the notes of table 2 should be part of the text (maybe in the discussion section) because they represent valuable data.
  9. As said before, it is not clear how you obtained the data in table 3.
  10. Lines 199-200: why do you give this kind of explanation only for this result? In my opinion, or you describe in general how to interpret regression results or provide the same description for each regression.
  11. Did you adjust your model for all the variable all together? In the text, do you report the result for the unadjusted or the adjusted model? In my opinion, you should describe both and highlight the differences between the two models (tables 4 and 5).
  12. Also for table 4 comments, please, respect the order of the parameters listed in the table.
  13. Did you try the stratification by sex to see if there any differences between males and females?

DISCUSSION

  1. Please, should consider a revision on the entire discussion organization: first, give a very brief explanation of your main results, then discuss your results in the order they appear in the result section. In my opinion, now the discussion section is focusing. You start describing the last data of your results without making any comparison with previous literature. After the explanation of other results, you turned again, speaking about regression results.
  2. Lines 236-237: I do not understand the meaning of this sentence.
  3. Is it essential to explain diet soda if the result is not significant?
  4. Lines 240 – 245: this paragraph is complicated in understanding. On which basis did you make your suppositions?
  5. Lines 246 – 254: in this paragraph, you speak about data not reported or not highlighted in the result discussion. This part best fits in the results section. Please add comments to these data in the discussion section.
  6. The second study limitation is unclear.
  7. In my opinion, the implications contents do not fit with this kind of section.
  8. In my opinion, in the conclusion section, you should go directly to your conclusion statements, avoiding repeating what your study is about and involving. You are saying it later in the conclusion explanation.

Reviewer 3 Report

Nicely written article and well explained. Some minor points to clarify below:

2.2.
Inclusion/exclusion criteria

2.3.1.
"Both TSRQs for diet and exercise were translated from English to Dutch by the researchers."
It seems that the English version was not validated for the Netherlands. Simply translation cannot reflect true meaning.

2.3.5.
ICC - which model?

Reviewer 4 Report

Review of the work entitled “Understanding vocational students' motivation for 3 dietary and physical activity behaviors”. The work concerns the relationship between various types of motivation and selected health behaviours within the context of the theory of self-regulation. It is a coherent study, cognitively interesting, but also in terms of application within the context of planning health education and health promotion programmes. I am also positive about the critical and reliable preparation of section 4.1. (‘Limitations’).

Comments
Ad 2.1. Would it be possible (even briefly) to characterise the aforementioned programme in the area of youth health promotion?
Ad 2.3.2. I wonder, what the criteria for considering only the presented aspects in the analysis of eating behaviours were? Other food choices (e.g. consuming vegetables, fish, etc.) seem to be equally diagnostic. The limited number of nutritional variables analysed is a limiting factor of the work (which should be indicated in section 4.1.).
Ad. 2.3.4. The assessment of BMI on the basis of the declared values of body mass and height cannot be regarded as objective (and thus, constitutes a limitation of the work).

Round 2

Reviewer 2 Report

The authors improved the manuscript quality, and some previously unclear sections are now clear.